# New Perspectives on the Role of Liquid Biopsy in Bladder Cancer: Applicability to Precision Medicine

**DOI:** 10.3390/cancers16040803

**Published:** 2024-02-16

**Authors:** Fernardo Alberca-del Arco, Daniel Prieto-Cuadra, Rocio Santos-Perez de la Blanca, Felipe Sáez-Barranquero, Elisa Matas-Rico, Bernardo Herrera-Imbroda

**Affiliations:** 1Departamento de Urología, Hospital Universitario Virgen de la Victoria (HUVV), 29010 Málaga, Spain; fernando.alberca@ibima.eu (F.A.-d.A.); rocio.santosperez.sspa@juntadeandalucia.es (R.S.-P.d.l.B.); felipe.saez.sspa@juntadeandalucia.es (F.S.-B.); 2Instituto de Investigación Biomédica de Málaga y Plataforma en Nanomedicina (IBIMA Plataforma BIONAND), 29590 Málaga, Spain; 3Departamento de Anatomía Patológica, Hospital Universitario Virgen de la Victoria (HUVV), 29010 Málaga, Spain; juand.prieto.sspa@juntadeandalucia.es; 4Unidad de Gestion Clinica de Anatomia Patologica, IBIMA, Hospital Universitario Virgen de la Victoria, 29010 Málaga, Spain; 5SYNLAB Pathology, 29007 Málaga, Spain; 6Genitourinary Alliance for Research and Development (GUARD Consortium), 29071 Málaga, Spain; 7Departamento de Biología Celular, Genética y Fisiología, Universidad de Málaga (UMA), 29071 Málaga, Spain; 8Departamento de Especialidades Quirúrgicas, Bioquímica e Inmunología, Universidad de Málaga (UMA), 29071 Málaga, Spain

**Keywords:** liquid biopsy, bladder cancer, immunotherapy, precision medicine

## Abstract

**Simple Summary:**

Due to the high heterogeneity and variability of bladder cancer, the identification of new biomarkers involved in the development and mechanisms of tumor progression and resistance to treatment offers the possibility of adapting our treatment options based on the molecular characteristics of each specific tumor subtype. In recent years, liquid biopsy has emerged as an innovative technique to analyze unique tumor components released into the peripheral circulation in body fluids such as blood and urine. This review presents the most recent advances in researching the applicability of liquid biopsy in bladder cancer, as well as an update on the most commonly used biomarkers and new techniques for their evaluation. The advantages and disadvantages of these methods are outlined based on the recent literature on the subject.

**Abstract:**

Bladder cancer (BC) is one of the most common tumors in the world. Cystoscopy and tissue biopsy are the standard methods in screening and early diagnosis of suspicious bladder lesions. However, they are invasive procedures that may cause pain and infectious complications. Considering the limitations of both procedures, and the recurrence and resistance to BC treatment, it is necessary to develop a new non-invasive methodology for early diagnosis and multiple evaluations in patients under follow-up for bladder cancer. In recent years, liquid biopsy has proven to be a very useful diagnostic tool for the detection of tumor biomarkers. This non-invasive technique makes it possible to analyze single tumor components released into the peripheral circulation and to monitor tumor progression. Numerous biomarkers are being studied and interesting clinical applications for these in BC are being presented, with promising results in early diagnosis, detection of microscopic disease, and prediction of recurrence and response to treatment.

## 1. Introduction

Bladder cancer is the ninth most common cancer worldwide and the tumor most diagnosed in the urinary tract. Its incidence is increasing, and it is particularly high in Western Europe and North America [1]. The main histological subtype is urothelial carcinoma (uBC), which is diagnosed up to 90% of the time. Other subtypes are squamous carcinoma (SCC) (5%), adenocarcinoma (0.5–2%), and small-cell carcinoma (<1%) [2]. 

A histological classification system for UCs, which included papillary urothelial neoplasm of low malignant potential (PUNLMP) and non-invasive papillary carcinoma low-grade (LG) and high-grade (HG), was published by the World Health Organization (WHO) in 2004. Then, this system was incorporated into the revised WHO classifications for 2016–2022. It replaced the previously used 1973 WHO classification, which simply differentiated grades one (G1), two (G2), and three (G3). However, a downside of the change was that the WHO 1973 classification was more effective in predicting the development of TaT1 NMIBC compared to the WHO 2004/2016 classification. Furthermore, they were similarly limited in that the WHO 1973 and the WHO 2004/2016 were both prognostic for progression but not for recurrence. Today, a combination (LG/G1, LG/G2, HG/G2, and HG/G3) of both classification methods is used, which offers a more detailed method of patient stratification among specific categories and has demonstrated its superiority over either individual classification system.

According to the TNM classification, BC is divided into two groups: non-muscle-invasive bladder cancer (NMIBC) and muscle-invasive bladder cancer (MIBC). In BC patients, almost 75% are diagnosed in the non-muscle-invasive cancer (NMIBC) stage and can then eventually progress to muscle-invasive bladder cancer (MIBC). The standard therapeutic method in NMIBC is transurethral bladder resection (TURB) followed by intravesical chemotherapeutic drugs (mitomycin-C, epirubicin, pirarubicin, or gemcitabine) or a bacteria-based immunotherapy called Bacillus Calmette–Guérin (BCG), depending on the risk [3]. Meanwhile, due to high rate of recurrence and progression in the muscle-invasive stage, particularly in high-risk tumors, patients require close follow-up with multiple cystoscopies and urine cytology, which entails a high economic cost for the health care system.

Currently, cystoscopy and tumor tissue biopsy remain the gold-standard diagnostic tools, although they are still invasive and can lead to pain and infective complications [4]. In addition, other disadvantages of taking this approach, such as sampling bias or difficulty of sampling deep tumors, limit its use in large-scale screening. Meanwhile, less invasive techniques like urine cytology (has high sensitivity in HG and G3 tumors (84%) but low sensitivity in LG/G1 tumors (16%)) and ultrasound are limited in their diagnostic potential, especially in low-grade tumors [5]. Furthermore, other imaging modalities such as computed tomography (CT) scans or magnetic resonance imaging (MRI) may have certain limitations in their clinical application due to the potential risks associated with ionizing radiation exposure, the high costs involved, and the delay often involved in identifying recurrence or metastasis [6,7]. Today, the poor sensitivity and other limitations of cytology and the complications associated with invasive techniques greatly motivate clinicians to find new diagnostic, prognostic, and predictive methods, though doing so presents a serious challenge. In one avenue of research development, it is proposed that the identification of urinary biomarkers could improve and enhance the diagnosis and screening techniques and enable us to determine a more accurate recurrence rate.

Furthermore, in the context of MIBC and/or advanced/metastatic BC, certain clinicopathological features have demonstrated prognostic significance. These include gender, urothelial tumor subtype, presence of lymphovascular invasion, involvement of prostatic urethra, previous radiotherapy, TNM stage, local recurrence, nodal involvement, distant metastasis, previous chemotherapy, and perineural invasion and death.

The standard of care for MIBC is radical cystectomy. If the patient undergoing this procedure is suitable, the addition of neoadjuvant platinum-based chemotherapy has been shown to improve complete response rates at surgery and overall survival (6–8% at 5 years). Alternatively, for patients who are not candidates for radical cystectomy due to comorbidities, there is the option of performing a bladder preservation protocol with TUR + radiotherapy +/− chemotherapy. However, clinical outcomes are poor and MIBC patients progress to receive other treatments as their cancer spreads to the lymph nodes and other sites, and approximately half of these patients develop metastases and die within 3 years [8].

In this context, optimizing medical treatment and reducing morbidity are major goals [9]. Great potential is held by immunotherapy, and particularly the use of immune checkpoint inhibitors (ICIs) targeting the programmed death 1 (PD-1)/programmed death ligand 1 (PD-L1) axis, which confers benefits for the treatment of different forms of UBC. ICIs targeting PD-1 (pembrolizumab) or PD-L1 (atezolizumab) have been approved as first-line therapy of metastatic MIBC platin-ineligible patients or as second-line therapy for patients with metastatic urothelial bladder carcinoma (mUC). Moreover, nivolumab (anti-PD-L1/2) has recently been approved for use in high-risk non-metastatic advanced MIBC (pT3-4 or N+), and avelumab (anti-PD-L1) has been shown to facilitate 14–21 months of maintenance after a good response to first-line metastatic MIBC treatment [10]. However, despite the promising results in 20–35% of patients with advanced or recurrent BC, the overall survival remains limited, indicating the complexity of tumor-induced immune alteration. In that context, there is an urgent need to understand the mechanisms of tumor immune evasion and resistance to immunotherapies. This will have the added advantage of opening up opportunities for developing new therapeutic approaches that reduce recurrences and progression and enhance patient outcomes.

Today, driven by ambitions to overcome the limitations of the current diagnostic methods, new approaches are emerging to improve our understanding of the molecular landscape of BC. For instance, in recent years, liquid biopsies have enabled the generation of genomic profiles from various extracted body fluids (such as blood, urine, saliva, cerebrospinal, and pleural fluids), which are tested for a strong correlation with the genomic profiles of tumors [11,12]. Liquid biopsy is the term used to describe the analysis of biomarkers in body fluids. It is becoming important for determining the genomic landscape of cancer patients, assessing their treatment responses, quantifying whether there is minimal residual disease, and evaluating therapeutic resistance.

This method avoids invasive procedures while yielding molecular information comparable to that obtained from traditional tissue biopsies [13,14]. Since liquid biopsy serves as a non-invasive technique, it enables periodic monitoring of tumor biomarkers. This monitoring facilitates the assessment of tumor progression, aids in the selection of personalized treatment strategies, and provides valuable information on therapeutic responses. Additionally, liquid biopsy offers the ability to collect different molecules like circulating tumor cells (CTCs), circulating cell-free tumor DNA (ctDNA), circulating cell-free tumor RNA (ctRNA), proteins, peptides, and metabolites. These components can be extracted from a single sample and employed in various tests (see Figure 1) [15,16]. 

Furthermore, liquid biopsy offers an added advantage in mitigating or eliminating false information caused by disparities within the tumor, thereby overcoming the variability associated with genomic information derived from tissue analysis, which can depend on the location and accessibility of the tumor [17]. Various lines of evidence suggest that tumor heterogeneity may play a role in the diverse clinical outcomes observed in MIBC and NMIBC, and several studies have compared results between conventional urine cytology and new tests based in liquid biopsy, generally finding that the new tests are more sensitive than cytology and have greater specificity [18,19]. 

Liquid biopsy has produced novel findings already. For instance, mutations in DNA damage repair genes [20,21] and somatic mutations in tyrosine-protein kinase receptor 2 (ERBB2) [22] have been linked to excellent responses to neoadjuvant chemotherapy (NAC) in patients with muscle-invasive bladder cancer (MIBC). Meanwhile, gain-of-function mutations in fibroblast growth factor receptor 3 (FGFR3) have been shown to be prevalent in low-grade non-muscle-invasive bladder cancer (NMIBC). The liquid biopsy approach thus presents a promising alternative to tissue biopsy for disease diagnosis, monitoring tumor progression, and assessing the treatment response [23].

Urine is the most advantageous option for BC liquid biopsy applications. This is primarily attributed to its constant contact with the bladder mucosa and tumor, ease of acquisition, minimal patient compliance requirements, and lower contamination compared to blood, despite blood being the most commonly described and utilized fluid in liquid biopsy for various cancer types. In BC diagnostics, owing to its high abundance of tumor marker proteins and the strong correlation between markers and diseases, urine emerges as a particularly valuable choice [24]. Then, in advanced BC stages, blood-based biomarkers or liquid biopsies may be especially valuable as they offer a strategy to detect minimal residual disease (MRD) earlier than conventional imaging tests [25]. 

In sum, for all the reasons described here and more, the landscape of precision medicine in oncology is changing towards liquid biopsy. A multitude of studies have underscored the significant role of this non-invasive test in the comprehensive management of patients with BC across various stages [26]. To draw those together, our objective when conducting this review was to present the most recent findings about the different components of liquid biopsy and their clinical value in diagnosis, early recurrence detection, and individualized treatment selection.

## 2. Materials and Methods

This is a narrative review based on a MEDLINE and PubMed search for contributions in the last three years (between January 2020 and 17 September 2023) using the keywords “liquid biopsy”, “bladder cancer”, “immunotherapy”, “molecular landscape”, “tumor heterogeneity”, and “precision medicine”, which were sought within the title and/or abstract.

Selection criteria were the publication date, study type, publication source, technological approach, clinical outcomes, and a focus on liquid biopsy.

First, we conducted a search for articles about liquid biopsy in bladder cancer. We identified a total of 112 studies, 24 narrative reviews, 2 systematic reviews (though 1 was excluded because it did not meet our selection criteria), and 1 multicenter study.

After a careful review of which to include in this article, we selected those that exposed the novelties and latest advances of liquid biopsy or described its current status of application in bladder cancer, both in the urine and blood.

## 3. Tumor Heterogeneity in Advanced BC

MIBC has a significant degree of heterogeneity, manifesting as pronounced genomic instability and a heightened frequency of somatic mutations, like in lung or pancreatic cancer. This poses a challenge in the management of advanced disease [27]. The presence of heterogeneity can be discerned across individuals (inter-patient heterogeneity), or within an individual either in the form of intra-tumoral heterogeneity (among primary tumor cells) or inter-tumoral heterogeneity (between primary or metastatic sites) [28]. Exploring the effects of heterogeneity at the individual level across different molecular subtypes has resulted in significant insights into the genomic landscape of MIBC. For instance, an examination of 178 cancer-related genes in a cohort of 110 patients with MIBC revealed that ERBB2 missense mutations are only observed in patients who respond positively to NAC [22]. However, most of the mutations identified in these multifocal tumors are limited to isolated fragments, suggesting that these findings may not accurately reflect the full panel of the disease. Securing comprehensive knowledge represents a challenge in accurately characterizing the genomic profiles of patients affected by this disease [29].

Advances in scientific technology such as next-generation sequencing (NGS) techniques have allowed for improved description of tumoral heterogeneity at different levels. NGS has been performed in several urological tumors such as renal cancer [30], prostate cancer [31], and bladder cancer [29], making it possible to understand why treatment failures occur and to investigate the mechanisms of resistance in these pathologies. Although these techniques are not currently used in routine clinical practice, the development of this field of research opens the door to new models of targeted therapy applicable to precision medicine.

Based on novel findings, different classifications of MIBC subtypes have been put forward in recent years, highlighting the importance of the progress that is being made for the clinical stratification of patients and their responses to treatments. Based on these molecular classifications for MIBC, diverse studies have been carried out on immune marker expression in the different subtypes. The updated Cancer Genome Atlas (TCGA) reported on five mRNA-expression-based molecular subtypes: luminal–papillary, luminal–infiltrated, luminal, basal–squamous, and neuronal [32]. In total, 412 MIBCs were identified in the two main groups: luminal and basal–squamous. Each subtype was associated with distinct mutational profiles, histopathological features, and prognostic and treatment implications. In general, research has since shown better results for the luminal subtypes, which are more greatly infiltrated by immune cells (due to higher expression of immune biomarkers such as PD-1 and PDL-1), while the neuronal subtype has been found to have the worst prognosis [33,34]. As for the basal/squamous subtypes, increased expression of immune control markers and epidermal growth factor receptor (EGFR) has been observed, and research has shown that it may result in a better response to targeted therapies. Through such findings, these studies suggest that the response to treatments used in advanced bladder cancer may be influenced by the genomic and molecular characteristics of each specific MIBC subtype. Hence, genomic analyses of a single tumor biopsy or specimen could underestimate the mutational burden of heterogenous BC.

## 4. Liquid Biopsy: A New Non-Invasive Monitoring Test in BC

Molecular stratification of tumors is coming to play a crucial role in assessing the efficacy and applicability of targeted treatments and immunotherapeutic approaches that are being developed in BC. In this regard, given that there can be substantial inter-tumor heterogeneity, it is important to acknowledge that analyzing a solitary sample from a specific place may not accurately reflect the patient’s overall cancer profile. To address this issue, liquid biopsy has emerged as a novel non-invasive methodology for conducting a comprehensive and dynamic assessment of molecular markers that are secreted by tumors [15].

In recent years, there have been significant advancements in the field of BC therapy, leading to a better understanding of molecular profiles. For instance, some studies have demonstrated the dual function of plasma cells (PCs) in both driving and controlling cancer development [35]. A recent study has confirmed the prognostic impact and efficacy of ICIs, based on crosstalk between the major functional subtypes of plasma cells (IgG1 and IgA1) and tumor cells. This study establishes that understanding this mechanism is crucial for predicting clinical approaches’ influence on survival rates and response to ICIs [36]. Despite cystoscopy being the most sensitive approach for diagnosing BC, its rate of complications, high costs, and invasive nature have inspired efforts to develop alternatives to this gold standard, ones that can match its sensitivity but without such invasiveness and complications and that also demonstrate greater cost-effectiveness. In this context, the cost of liquid biopsy analysis, involving sample collection, processing, and specialist laboratory techniques, could limit its extensive use. Likewise, methods like next-generation sequencing (NGS) can be costly and need specialist infrastructure and training. The need for regular testing and ongoing monitoring may also increase overall expenses. For liquid biopsies to be valuable in routine clinical practice, it is necessary to achieve a balance between cost-effectiveness and therapeutic efficacy [37].

### 4.1. Diagnostic Potential of Circulating Tumor Cells, Circulating Non-Coding RNAs (ncRNAs), and Cell-Free DNA

In a recent study, Kouba et al. provided a comprehensive review of the clinical applications of various types of biomarkers detectable by liquid biopsy. Tumor-derived cell-free DNA has presented a significant breakthrough in molecular diagnostics, with potential clinical applications that provide solutions that were previously enigmatic [38]. By using advanced technology to investigate a blood sample, we can isolate tumor-derived material from circulating tumor cells (CTCs), circulating nucleic acids such as cell-free DNA (cfNDA), and non-coding RNA (ncRNA) for metabolomics. Accordingly, recent advances in omics technologies, including genomics, epigenomics, proteomics, transcriptomics, and metabolomics, have improved our understanding of the molecular landscape causing cancer [39]. Although CTCs must be obtained from blood samples, cfDNa and miRNA can be analyzed in blood or urine.

#### 4.1.1. CTCs

CTCs are tumor cells secreted into the peripheral blood circulation with intact nuclei, characterized by cytokeratin (CK)-positive and CD45-negative markers. These cells play an important role in the hematogenous dissemination of tumors [40]. Currently, there are several methods for the detection and analysis of CTCs. The most commonly used methods are reverse-transcriptase polymerase chain reaction, immunochemistry, flow cytometry, and the CellSearch system. CellSearch is the only platform approved by the USA Food and Drug Administration (FDA) [41]. The CellSearch kit comprises a sample preparation kit and the CellSearch epithelial cell kit, which is used to concentrate cells expressing the epithelial cell adhesion molecule (EpCAM). Once isolated, individual cells are stained with the nucleic acid 40, 6-diamidino-2-phenylindole (DAPI), as well as with labeled monoclonal antibodies that specifically target leukocytes (CD45-allophycocyanin) and epithelial cells (cytokeratin 8,18,19-phycoerythrin). Then, a semi-automated fluorescence microscopy device allows computer-generated images to be generated [42].

BC, like most solid tumors, is epithelial in nature; therefore, the most commonly used marker for CTCs is EpCAM [43], which is predominantly expressed in high-risk tumors. In addition, one study identified EpCAM as an imaging marker for metastatic lymphatic disease, suggesting that it may be a sensitive and robust marker for detecting affected lymph nodes and promoting an increased rate of pathological lymph detection after lymphadenectomy [44].

In many solid tumors, in several types of cancer and at various stages of disease, an elevated CTC count has been associated with a worse prognosis, especially in lung, breast, gastric, and colorectal cancer. However, in BC, CTCs have generally appeared to be a poor prognostic marker. Nonetheless, recent studies have observed that CTC density seems to be associated with earlier disease recurrence and shorter overall survival (OS). Therefore, the predictive value of CTCs in BC should be considered [45,46,47,48].

Additionally, some authors have investigated a relationship between CTCs and advanced tumor stage, lymph node involvement, and OS. These findings provide evidence that CTCs can serve as a reliable indicator of disease progression [49,50], and that the presence of CTCs may be used as a significant independent predictor of disease-free survival (DFS). Furthermore, one study found that the number of CTCs was higher in the MIBC group compared with the NMIBC group [51].

CTCs have been also investigated as biomarkers of treatment response. In one such study, the authors revealed that MIBC patients treated with previous NAC had a lower CTC density, and patients who had higher concentrations of CTCs benefitted more from NAC treatment [50,52]. Furthermore, in relation to immunotherapy treatment, recent studies have proposed that PD-L1 assessment via CTC monitoring could be a useful tool to apply to optimize treatment with PD-L1 inhibitors in “non-responder” BCG patients [53]. Elsewhere in the literature, in a prospective study carried out in non-metastatic BC patients after cystectomy, a research team analyzed the human epidermal growth factor receptor 2 (HER-2) expression in CTCs, showing that the HER2-positive CTC group could benefit from targeted therapies; however, HER-2-negative CTCs conferred resistance against immunotherapy to patients with HER-2-positive primary tumors. Additionally, preoperative CTCs were a potential predictor of early disease recurrence and cancer-specific and overall mortality [54]. Finally, in regard to the treatment response, it was shown that the efficacy of immunotherapies including checkpoint inhibitors, which are active in a subset of patients with positive CTCs and often induce durable responses, could be assessed by serial whole-blood CTC assays [55].

A novel technique based on the detection of CTCs is CytoTrack, an “open” manual research system that allows for studying various markers for CTC enumeration and characterization. Although it has yet not been clinically validated, CytoTrack has been proven to have a recovery and reproducibility matching that of CellSearch, which is promising. CytoTrack may thus have the potential to be used in future studies including tests of CTCs in clinical samples from breast, colon, and prostate cancer patients. Moreover, further markers, both surface and intracellular phenotypic and genetic markers, can be explored to improve the performance of CytoTrack across different cell types. In comparative in vitro studies with the CellSearch system, the results for CTC number (total number of both single cells and cells in clusters) showed similar recoveries by CytoTrack and CellSearch. CellSearch, however, had a lower variability in the analysis, which may have been due to the automated procedures versus the manual procedures in the CytoTrack analysis [56]. Looking to the future, to improve on the current options, several studies are investigating novel techniques such as nanotechnology-based techniques and microfluid circuits [52].

These studies to date collectively demonstrate that the detection and analysis of CTCs in the context of bladder cancer have numerous clinical applications in the field of precision medicine. Specifically, in prognosis and staging, the presence of CTCs in the bloodstream can offer insights into the aggressiveness of bladder cancer and its potential for metastasis, and a higher number of CTCs may be associated with a worse prognosis. Following that, continuous monitoring of CTC levels during treatment then proves valuable in evaluating the effectiveness of therapies. A reduction in the number of CTCs may indicate a positive response to treatment, whereas an increase could indicate resistance or progression. Similarly, the presence of CTCs post-treatment may indicate the persistence of resistant tumor cells.

Furthermore, the identification of specific genetic mutations and molecular profiles of CTCs holds the potential to inform the selection of targeted therapies or immunotherapies that are more effective than others against the specific characteristics of a patient’s bladder cancer. Moreover, in terms of early detection of recurrence, CTCs could help to identify recurrences before they become evident through other tests. This early detection capability enables timely (and thus potentially more effective) interventions, emphasizing the crucial role of CTC analysis in enhancing the precision of bladder cancer management.

Although methods that focus solely on the detection of CTCs have limited diagnostic value at present, new technologies may eventually make CTCs applicable as a predictive and monitoring tool in BC.

#### 4.1.2. Circulating Non-Coding RNAs (ncRNAs)

ncRNAs have been found to be regulators of internal cell signals that control various levels of gene expression, playing a vital role in maintaining cellular homeostasis and cell deregulation [57]. The ncRNA family is extensive, including transfer RNAs (tRNA), ribosomal RNAs (rRNAs), small nucleolar RNAs (snoRNAs), long non-coding RNAs (lncRNAs), microRNA (miRNAs), and circular RNA (circRNA). These types of ncRNAs are currently the most investigated in bladder cancer (BC). miRNA is the most extensively studied ncRNA of them all, but the roles of circRNA and lncRNA in tumor development and progression are increasing, sparking interest in these as biomarkers in liquid biopsy.

circRNA consists of RNA fragments without 5′ and 3′ ends or a poly(A) tail. This unique structure provides circRNA with high stability and resistance to nuclease degradation. Analysis of circDNA (circular DNA) in liquid biopsy is an interesting and promising area of research in bladder cancer. Some implications of circDNA in liquid biopsy in bladder cancer are its important role in diagnosis and prognosis, tumor burden monitoring, identifying mutations and genetic alterations, monitoring the response to treatment, and detecting treatment resistance. In particular, the presence of certain biomarkers in circDNA may have major prognostic implications, helping to predict cancer progression and patient survival. In addition, circDNA-based liquid biopsy offers a non-invasive option for sampling and follow-up over time, avoiding the need for repetitive tissue biopsies. Furthermore, circRNA has been implicated in the epithelial–mesenquimal transition process, resulting in several cell alterations such as increased invasive capabilities [58]. Certain authors have identified circRNA-targeting miRNA (such as circPRMT5, circMTO1, circTFRC, circBIRC6, circFLNA, etc.) that affect BC cell migration and invasion through the EMT process [59]. These findings underscore the importance of circRNA in mechanisms underlying lymphatic metastasis in BC [60,61,62]. Future studies and technological advancements may aid in comprehensively harnessing the potential use of this biomarker in BC.

lncRNAs, meanwhile, are RNA transcripts that exceed 200 nucleotides in length and do not encode proteins. lncRNAs regulate numerous cancer-related biological processes and have an important influence on transcriptional regulation, epigenetics, cancer progression and metastasis, and response to treatment [63]. Specifically, recent studies have shown that lncRNAs have a role in several cell processes such as cell proliferation, apoptosis, angiogenesis, migration, and the spread of cancer cells [64]. The molecular processes of lncRNAs in BC have been studied, including lncRNAs interacting with DNA, RNA, and proteins. lncRNAs usually have limited expression and lower evolutionary conservation but they may be found in higher amounts than normal in the urine and plasma of individuals with BC, which could be useful for developing novel diagnostic or prognostic tools. Furthermore, it has been reported that abnormal expression of thirty-six lncRNAs is closely linked to several clinical features of BC. One study looked at using thirty lncRNAs that can be up-regulated as possible clinical indicators of BC. Among the fifteen lncRNAs that were found to be more active in bladder cancer, their analysis revealed that UCA1 had a reasonably high level of accuracy, sensitivity, and AUC, meaning it may represent the most promising diagnostic biomarker. 

*UCA1* was the first known cancer-causing long non-coding RNA and is greatly overexpressed in BC. It helps promote BC development by controlling several targets and pathways [65]. To start with, *UCA1* disrupts the chromatin remodeling function of BRG1 and attaches to the P21 promoter, causing the growth of cancer cells. The activation of *UCA1* through C/EBPα also helps increase the survival and decrease the cell death of BC cells [66]. Additionally, *UCA1* controls the expression of miR-16/GLS2 and inhibits the generation of ROS. Through the mTOR/STAT3 pathway and the miR143/HK2 axis, *UCA1* also boosts glucose metabolism in cancer cells [67]. Additionally, UCA1 was shown to affect *AKT* expression and activity. Changes in *UCA1* levels were seen to be like the changes in *CREB* expression and phosphorylation, which play a role in promoting cell proliferation and regulating the cell cycle [68]. In this case, *BMP9* enhances the levels of AKT phosphorylation and promotes the production of UCA1, to enhance the proliferation and metastasis of BC cells [69]. Beyond UCA1, additional lncRNAs have been discovered in the urine exosomes of individuals with high-grade MIBC. These include *HOTAIR*, *HOX-AS-2*, *MALAT1*, *HYMAI*, *LINC00477*, *LOC100506688*, and *OTX2-AS1* [64].

While the roles of circRNA and lncRNA in tumor development and progression are increasing, miRNA remains the most extensively studied ncRNA. miRNAs form a class of small ncRNAs that typically consist of approximately 22 nucleotides. They perform regulatory control of gene expression by binding to the 3’untranslated region of their target messenger RNA (mRNA). miRNAs have a significant role in post-transcriptional regulation, proliferation/apoptosis (some microRNAs may have implications for bladder cancer formation and progression), and immunity. Accordingly, the differential expression of microRNAs has been proposed as a potential biomarker for the diagnosis and prognosis of bladder cancer. In urine, there is an abundant presence of unbound nucleic acids in the sediments, which makes urine a valuable medium for the detection of miRNA [70].

A multitude of research has been conducted to investigate the diagnostic and prognostic capabilities of miRNA biomarkers in BC. In this review, we provide the findings of some particularly interesting research. For instance, Feng et al. investigated the involvement of miR-19a and miR-99a in BC. Their findings revealed that, like the pattern observed in cancer tissue, miR-19a is also up-regulated in plasma, acting like an oncogene, and miR-99a is down-regulated [71,72]. Similarly, another study showed that miR-200b is overexpressed in MIBC. Inversely, miR-33b and miR-92b demonstrate reduced expression levels and show a negative correlation with the pathological stage of the disease [73]. The findings over the last ten years have encouraged the creation of panels that include numerous miRNAs for pooled analysis. One study employed a panel of 74 miRNAs and discovered that 33 miRNAs were elevated and 41 were down-regulated in urothelial cancer patients compared to controls, with the most relevant being let-7miR, miR-196a, miR-1268, miR-143, miR-100, miR-101, miR-1, and miR-200 [15]. Furthermore, through targeting zinc finger E-box 1,2 (*ZEB1*, *ZEB2*) and epidermal growth factor receptor (*EGFR*) binding sites, miR-200 has been identified as a regulator of the epithelial–mesenchymal transition in malignant cells [74]. A recent investigation discovered greater amounts of miRNA-373 in the plasma of BC patients than in healthy controls, and its overexpression was linked to enhanced tumor proliferation, migration, and invasion [75]. Elsewhere, several authors established the predictive significance of plasma miR92-a, miR-100, or miR-143 in BC patients, with high sensitivity and specificity. They also discovered that the more advanced the stage, the higher the amount of plasma miR-210, while there was a notably lower level in patients who received surgery [76]. Researchers have also put forward miR-205 as a possible biomarker for distinguishing NMIBC from NMIBC [77].

A recent investigation in patients with BC discovered that miR-96 and miR-210 were present in urine sediment, although the control cystoscopy was negative [78]. In another study, it was proposed that a panel of 12 miRNAs could reduce cystoscopy rates by 30%, increase specificity and sensitivity, provide a higher diagnostic yield, and provide more accurate information on recurrence rate, aggressiveness, and stage than traditional diagnostic methods, thereby highlighting its suitability to become the preferred methodology in the future [79]. A study of miRNA profiles in urine, meanwhile, utilizing NGS-derived analytic tools, categorized multiple cancer subtypes as follows: up-regulation of miR-21-5p, miR-106b-3p, miR-486-5p, miR-151a-3p, miR-200c-3p, miR-185-5p, and miR-224-5p was observed in patients with non-invasive G1/G2 urothelial carcinoma, while down-regulation of miR-30c-2-5p and miR-10b-5p was observed in patients with G3 non-invasive urothelial carcinoma. As well as the categorization, these findings demonstrate that NGS analysis could offer the ideal approach to conducting biomarker research involving miRNAs [80]. When taking all findings presented here into consideration, in sum, the ability to detect miRNA panels could help us to diagnose and monitor the disease more accurately than cystoscopy.

#### 4.1.3. Circulating Tumor DNA (ctDNA)

Tumors release DNA into the peripheral circulation and urine, with cfDNA representing a significant proportion. As such, while most of the DNA in plasma originates from healthy cells, in cancer patients, a small portion comes from circulating tumor DNA (ctDNA). Genomic variations in ctDNA can be used as genetic signatures [81], and accordingly, the use of ctDNA in liquid biopsy for bladder cancer is a developing area of research and clinical application. The role of ctDNA in BC is particularly important in situations where obtaining tumor tissue through traditional biopsies is difficult. The applications of ctDNA in liquid biopsy for bladder cancer include the detection of mutations and genetic alterations, BC diagnosis, monitoring the response to treatment, early detection of recurrence, identification of prognostic biomarkers, and the possibility of conducting non-invasive and serial tests.

Specifically, epigenetic changes implicated in BC development and progression can be identified by studying ctDNA fragments released into the blood or urine [82]. DNA methylation stands out as one of the most significant epigenetic processes in BC. Normally, CpG dinucleotides in human DNA are methylated by DNA methyltransferases (DNMTs), which inhibits tumor development and cancer cell invasion by causing transcriptional silencing. However, hypomethylation may occur in cancer patients, and one study revealed extensive hypomethylation in NMIBC when compared to other cancers, lending support to the idea that DNA methylation plays a role in the carcinogenesis and aggressiveness of BC [83]. An additional finding to date is that some methylation genes (*SFRP1*, *FHIT*, *DCH1*, *PMF1*, *RUNX3*, *LAMC2*, or *RASSF1A*) are not just biomarkers but are also linked to worse clinical outcomes in cancer patients. Their evaluation has enabled predictive models to be developed for BC survival [84]. Numerous studies on the methylation of these genes have been conducted in recent years, with greater levels of methylation expressed in patient samples compared to healthy samples [85,86,87].

The alteration of histones is another epigenetic process that has recently been investigated in BC. Post-translational modifications (PTMs) of histones are involved in DNA repair, differentiation, and tumor development [88,89]. Some authors have investigated the relationship between these changes and taxonomic groups and confirmed that histone modification signatures represent the basal and luminal subtypes of BC; this is the case for H3K4me1, which has only been found to be present in basal tumors. The research findings indicate that changes in these histones may cause variability among MIBC subtypes [90]. The most current work in the field looked at how HSP90 inhibition affected protein expression and histone PTMs in BC. Five HSP90 inhibitors (AUY922, ganetespib, SNX2112, AT13387, and CUDC305) inhibited bladder cancer cell proliferation in a dose- and time-dependent form, demonstrating a link between proteomic changes and histone PTMs in response to HSP90 inhibitor treatment [91]. This supports the notion that PMTs and DNA methylation have the potential to be used as essential markers for targeted therapy. Such therapy may be effective as they are reversible, and inhibitory enzymes can correct their aberrant action and restore tumor suppressor gene expression [92].

Quantitative polymerase chain reaction (qPCR) and digital polymerase chain reaction (dPCR) are the most extensively used methods for detecting ctDNA and its changes in different body fluids. They are quick and inexpensive techniques, but their use is restricted as they can only monitor known genetic alterations [93]. To address this limitation, NGS-based methods have emerged and are increasingly being used to identify ctDNA in small panels as well as entire genome sequences, with whole-genome sequencing for kidney, bladder, and prostate cancer available since 2010. These innovative approaches do not require tumor tissue and can characterize newly customized genomic profiles. Their application to precision medicine might be critical in creating new personalized therapies, assessing the emergence of drug resistance, and quantifying minimal residual disease [94]. While several modalities for conducting data analysis using sequencing techniques have been presented, a consensus has not yet been reached on the modality that yields the most accurate outcomes. 

Although NSG has been employed in numerous BC diagnostic and management studies, few have focused on predicting outcomes for NMIBC patients [95]. One team conducted a surveillance trial of patients with NMIBC to predict progression using genomic variants in urinary and plasma ctDNA, discovering that patients with disease progression had substantially higher levels of tumor DNA in their plasma and urine than patients with disease recurrence. Furthermore, all patients with progressive disease, including those with no detectable ctDNA in their plasma, exhibited elevated urinary tumor DNA levels, whereas patients with recurrent NMIBC had lower levels [96,97].

Elsewhere, by applying NGS to urine cytology samples, another group investigated potential biomarkers that could predict a response to BCG in HR-NMIBC [98]. In their cohort of patients, *TERT* (61%), *KDM6A* (34%), *ARID1A* (27%), *KMT2D* (24%), *FGFR3* (20%), *CREBBP* (17%), and *EP300* (17%) were the most mutated genes. These mutations are comparable to those discovered through a TCGA project. The authors discovered that RMB10 and EPHA3 were statistically more prevalent in BCG-responsive patients, while *ARID1A*, *EP300*, and *CDKN1A* mutations tended to be more common in patients who did not respond to BCG treatment, but none of these differences were statistically significant.

In recent years, ctDNA determination in MIBC recurrence after RC has also been studied, with researchers noting that a higher proportion of ctDNA reflected a higher rate of persistence, demonstrating that it may be used as a predictor of recurrence [99]. Other studies have also found elevated levels in patients with metastasis [100]. Beyond those, a prospective study has offered a representative characterization of the genomic panel that may be used in liquid biopsy studies, based on findings from the tumor tissues of patients with BC. In addition, it has been demonstrated that NGS analysis of genes such as *FGFR3*, *ERBB2*, *ERCC2*, and *TMB* (proposed treatment response markers in simple tissue) offers a cost-effective and minimally invasive strategy for stratifying patients in clinical trials, which constitutes a new line of application in the field of precision medicine in cancer [101].

Finally, when we consider the importance today of immunotherapy in the treatment of patients with BC, ongoing analysis of ctDNA may allow for close surveillance during ICI therapy. Recent studies have analyzed its use as a marker in adjuvant immunotherapy treatment, and changes in its frequency have identified non-responders to targeted therapy, demonstrating that ctDNA levels can be used to monitor treatment response and predict the most effective targeted therapeutic strategies [102,103]. In support of this, a recent clinical trial indicated that early on-treatment reduction in the frequency of ctDNA variant alleles (VAF) may be a valuable predictor of long-term immunotherapy benefit [104]. Patients with a reduction in VAF at 6 weeks exhibited a substantially greater decrease in tumor volume, as well as prolonged progression-free and overall survival. Additionally, changes in genes like *TP53*, *TERT*, and *BRCA1*/*BCRA2* detected before and after treatment with new ICIs were shown to have a direct relationship with resistance in advanced BC.

### 4.2. Modern Biomarkers in BC: Metabolomics and Proteomics in Liquid Biopsy

The metabolome emerges from the interplay of the genome, epigenome, transcriptome, proteome, and external influences. Modern metabolomics technologies have led to notable progress in characterizing and distinguishing various cancers, as well as identifying potential targets for therapeutic interventions. Metabolomics seeks to leverage the metabolic profile of cancer when assessing disease risk, enabling real-time description and monitoring of disease stages, as well as producing great insights into cancer staging [105]. Presently, metabolomics is also acknowledged as a highly relevant approach to exploring individual phenotypes in cancer systems biology. In this regard, the discernible metabolic distinctions between malignancy and normal cells are widely recognized as a hallmark of cancer [106]. Among the most commonly employed techniques to test for these are gas chromatography–mass spectrometry (GC-MS) and liquid chromatography–mass spectrometry (LC-MS) [107]. Furthermore, despite lacking standardization in BC [104], nuclear magnetic resonance (NMR)-based methodologies offer an alternative for analyzing isolated biological samples without requiring pre-processing steps [108,109]. Moreover, recently, various separation techniques, including electrophoresis, reversed-phase liquid chromatography, ion exchange, and size-exclusion chromatography, as well as immune enrichment or depletion methods, have been employed to isolate or concentrate proteins or peptides. Following on from this, mass spectrometry, coupled with advanced software, has been utilized to precisely identify, characterize, and quantify proteomic data [110].

The potential for predicting the prognosis of urothelial carcinoma through metabolomic profiling of urine samples has been evidenced. In a recent study, urine samples from forty-one patients with transitional cell carcinoma (specifically, bladder cancer) and forty-eight healthy individuals were subjected to analysis using high-performance liquid chromatography–mass spectrometry methodology [111]. The outcomes revealed a sensitivity and specificity of one hundred percent. Furthermore, similar to metabolomics, microbiomic analysis has shown promise as a prognostic indicator in the context of urothelial carcinoma. Numerous studies have explored the relationship between microbial communities and clinical outcomes in patients with urothelial carcinoma. Extensive research indicates that microbial populations can influence urological conditions, suggesting a potential role of microbes across the spectrum of health and disease [112].

In sum, refinements in liquid biopsy techniques, including those focused on circulating biomarkers, can contribute to more accurate and non-invasive methods for diagnosing and monitoring bladder cancer. Such advancement could lead to earlier detection and improved surveillance of the disease.

## 5. Commercial Liquid Biopsy Kits for Bladder Cancer (BC) Screening and Follow-Up

Most research has been conducted on urine samples as opposed to blood samples. At the same time, advances in omics technologies, such as genomics, epigenomics, proteomics, transcriptomics, and metabolomics, have enhanced our comprehension of the intricate molecular factors underlying cancer [3]. The FDA has now approved six urinary diagnostic assays for BC: qualitative bladder tumor antigen (BTA) (*BTA stat*), quantitative BTA (*BTA TRAK*), quantitative nuclear matrix protein 22 (NMP22) (*Alere NMP22*), qualitative NMP22 (*BladderCheck*), fluorescent immunohistochemistry (*ImmunoCyt*), and fluorescence in situ hybridization (FISH) (*UroVysion*) for clinical use [113]. Three of them detect DNA, RNA, or protein alterations in urinary cells (UroVysion, BladderCheck, and ImmunoCyt), while another two quantify proteins released in the urine (NMP22 and BTA assays) [21].

### 5.1. Bladder Tumor Antigen Assay (BTA Test)

The BTA Stat and BTA TRAK tests are two laboratory procedures employing immunoassays to detect human complement factor H-related protein (hCFHrp) in urine samples obtained from individuals diagnosed with urothelial cancer [114,115], working on the basis that the expression of hCFHrp in the urine of bladder cancer patients is higher than in healthy ones. The distinction between the BTA Stat and BTA TRAK methods is that the BTA Stat test is quicker as it provides qualitative data, while BTA TRAK utilizes the enzyme-linked immunosorbent assay (ELISA) technique for quantitative analysis. The quantitative approach of BTA TRAK involves more time and employs a monoclonal antibody against hCFHrp. In clinical practice, the use of BTA TRAK alone to diagnose bladder cancer may result in unnecessary invasive interventions and financial burden. Conversely, using urine cytology alone runs the risk of overlooking a substantial number of bladder tumors, thereby raising the morbidity and mortality associated with advanced malignancy [116].

A meta-analysis of BTA Stat, which included 13 trials and 3462 patients, revealed that the test had higher sensitivity than cytology, especially in high-grade tumors, but less specificity. Specifically, these tests have high false positive rates, mainly attributed to the presence of blood in the urine samples examined [117]. Complement factor H is often found in the bloodstream, inevitably leading to incorrect positive results with the BTA Stat and TRAK assays in the presence of hematuria [118]. False positive results are more commonly observed with BTA Stat in comparison to the BTA TRAK test, occurring in less than 5% of individuals who do not have any known urine abnormalities [119].

### 5.2. Nuclear Matrix Protein 22 (NMP22)

Nuclear matrix proteins (NMPs) play a crucial role in inducing changes in various biological functions, including DNA replication and gene expression. These proteins are specific to mitosis and aid in the transfer of chromatids to descendant cells. In bladder cancer cell lines, the concentration of NMP22 can be up to 25 times higher than that found in normal urothelium, and importantly, NMP22 can still be detected after cell death [120]. Several studies conducted by different authors have compared the sensitivity and specificity of the NMP22 BladderChek test with that of cytology, and the findings suggest that urine cytology outperforms the NMP22 BladderChek, though performance is affected by factors such as leukocytes, current use of blood pressure control drugs, urinary calculi, creatinine levels, recurrent urinary tract infections, and hematuria [121]. A further finding was that combining both tests can enhance the sensitivity when diagnosing primary urothelial carcinoma, particularly in cases with deeper invasiveness (mostly > T2) and a higher tumor grade [122].

### 5.3. UroVysion

Human malignancies often exhibit chromosomal aberrations, and many solid tumors manifest complex alterations in genetic material. The UroVysion test employs multi-chromosomal fluorescence in situ hybridization (FISH) to detect aneuploidy, specifically involving chromosomes 3, 7, or 17. Additionally, a probe specific to the 9p21 locus is utilized in the analysis. This test demonstrates a high sensitivity for the detection of carcinomas and remarkable specificity in identifying in situ carcinomas as well as high-grade tumors [123]. In addition to its high specificity, this test is subject to little influence from other factors. The test may also be applied to assess for progression and recurrence. In a previous study, a positive U-FISH result was not associated with an increased risk of recurrence in the initial BCG treatment after TURB; however, a few months later, after BCG treatment (3 or 6 months), a positive test result was significantly correlated with the risk of tumor progression and recurrence (*p* < 0.001). The probability of recurrence was 3–5 times higher than in the negative group, and in comparison to the negative group, the rate of disease progression was 5–13 times greater. As such, a positive U-FISH following BCG perfusion was proposed to represent an independent risk factor for recurrence [124,125].

Furthermore, one study has shown that FISH has a prognostic role in predicting outcomes in individuals with NMIBC who have negative cystoscopy results and abnormal cytology. The decision not to perform a bladder biopsy in situations with atypical cytology and negative or inconclusive cystoscopy results, based on negative UroVysion results, has been shown to be cost-effective and has the potential to decrease avoidable adverse events [126].

### 5.4. ImmunoCyt

The ImmunoCyt test is an immunocytofluorescence-based assay that uses fluorescently labeled monoclonal antibodies to identify carcinoembryonic antigens and sulfated mucin glycoproteins expressed on most bladder cancer cells but not in normal cells. The sensitivity of the test varies significantly among different investigations and ranges from 60% to 100%, while its specificity ranges from 75% to 84% [127]. ImmunoCyt is an FDA-approved test used as a supplementary tool in the management of urothelial carcinoma, along with urine cytology and cystoscopy. ImmunoCyt had a superior pooled sensitivity of 72.5% compared to urine cytology, which had a sensitivity of 56.6%, as shown in a meta-analysis of seven investigations. Moreover, this test exhibits lower susceptibility to hematuria when compared to other tests, though it may be influenced by urinary infections, urolithiasis, and benign prostatic hyperplasia. Furthermore, ImmunoCyt had a lower specificity level of 65.7% in contrast to cytology’s specificity of 90.6% [128].

This cell-based biomarker appears to offer greater specificity and sensitivity in low-grade NMIBC compared to markers analyzing soluble tumor-associated antigens and urine cytology [129]. However, analyses have shown high interobserver heterogeneity, limiting the clinical application of this technology [130].

In the past decade, in addition to the FDA-approved kits described above, numerous other tests have been developed for the study and investigation of biomarkers in urine. The methods used for detection and surveillance of bladder cancer work based on DNA mutations or methylation, RNA signatures, improved cytology, and protein-based assays (Table 1).

### 5.5. New Biomarkers

Recent studies of new urinary biomarkers have aimed to enhance the diagnostic accuracy, sensitivity, and specificity of urothelial carcinoma diagnosis. These include BCLA-1 and BCLA-4 (two transcription factors present in the early stages of BC tissues, even before the appearance of a visible tumor), which are not influenced by infection, smoking, catheterization, or cystitis [147,148].

Aurora A kinase (AURKA), meanwhile, is a cell-cycle-associated kinase that regulates mitosis, is linked to BC progression, and correlates with stage, grade, and prognosis, particularly in patients with hematuria. Accordingly, the Aura Tek FDP Test^TM^ in urine distinguishes low-grade BC from normal urothelium and can detect the recurrence of BC in the presence of hematuria [149].

Another study demonstrated that the presence of elevated nicotinamide N-methyltransferase levels in urothelial carcinoma of the bladder can help determine the pathological grade [150]. Beyond that, apurinic/apyrimidinic endonuclease 1/redox factor-1 (APE/Ref-1) is an additional urinary biomarker that can be used to determine the stage, grade, and recurrence of bladder carcinoma [151]. Alternatively, as has been demonstrated in ovarian cancer, activated leukocyte cell adhesion molecule (ALCAM) can be used to determine the tumor stage and overall survival [152]. In addition, the UBC Rapid test, which detects elevated levels of cytokeratin 8 and 18, can help distinguish between patients with high-grade and low-grade urothelial cancer [153]. Another prospective study identified a panel of proteins capable of distinguishing MIBC from NMIBC by means of mass spectrometry. A panel of four sequenced polypeptides (uromodulin—UMOD, collagen alpha-1[I] chain—COL1A1, collagen alpha-1[III] chain—COL3A1, and membrane-associated progesterone receptor component 1—PGRMC1) predicted MIBC with a sensitivity of 81% and specificity of 57% [154].

In addition to evaluating proteins in urine, some studies have demonstrated the participation of proteins contained within urinary exosomes. These exosomes are small extracellular vesicles (EVs) that play an essential role in distinct stages of cancer development. These EVs are present in almost all human body fluids, including urine, and are notably rich in tumor exosomes in the microenvironment, making them a field of great interest and research for BC biomarkers [155]. Specific examples of proteins that are overexpressed in BC exosomes are alpha-1-antitrypsin (SERPIN1) and histone H2B type 1-K (HIST1H2BK), which were found at high concentrations in urinary EVs, with the presence of HIST1H2BK found to increase the risk of recurrence threefold [156].

Another study revealed that elevated periostin (POSTN)-rich EVs were associated with an increased risk of progression and a poor prognosis [157]. Furthermore, one of the most recent trials in this field of research obtained urine samples from the Biobank@UZA and the Belgian Virtual Tumor-Bank, from which EVs were isolated, and performed proteomic analysis using LC-MS followed by tandem mass spectrometry (MS). After enriching the EVs, the authors compared their results with those of previous studies, finding new potential biomarkers such as MASP2, C3, A2M, CHMP2A, and NHE-RF1. Additionally, HBB and HBA1 were identified as specific markers for recurrent BC. Then, in a recent study utilizing mass spectrometry, four proteins excreted in EVs (HEXB, S100SA4, SND1, and EHD4) were identified as potential BC biomarkers [158]. However, independent validation with a control group featuring other urological diseases is necessary to eliminate the possibility of nonspecific markers [159].

In another stream of work, a recent study identified urine-derived lymphocytes as a source of T cells in MIBC patients. CD8+ and CD4+ effector cells and regulatory T cells from urine precisely signaled the immune response of the body and provided a map of the tumor microenvironment (TME), which offers researchers the potential to determine the tumor’s stage and status [160]. For instance, a higher urine-derived lymphocyte count, such as high PD-1 expression on CD8+ prior to cystectomy, was associated with shorter recurrence-free survival. Today, urine-derived lymphocyte examination is being developed as a dynamic liquid biopsy that characterizes the immune TME and can be used to determine the prognosis of the disease [161].

These tests serve as tools for identifying BC, although cytology remains a considerably more cost-effective test. As such, the relevance of these advanced tests in terms of cost remains subject to debate. Today, no single test for hematuria screening has been established due to the limited cost-effectiveness of all options trialed, and it has not been conclusively proven to be superior to cystoscopy [162].

In Figure 2, we summarize the main liquid biopsy methods and their applicability in BC. As can be seen, each one holds some advantages over the other, and deeper investigations must be carried out to extend the current molecular findings in order to improve the diagnosis and prognosis of BC.

## 6. Conclusions

Precision medicine is developing rapidly in BC, and liquid biopsy has emerged as a new non-invasive diagnostic tool. BC has a high heterogeneity compared with other tumors, but the study of molecular alterations offers the possibility to perform a longitudinal and dynamic analysis throughout the disease, with benefits for diagnosis and prediction of recurrence, progression, or response to treatment. To date, liquid biopsy analysis has been most cost-effective in advanced stages of disease, especially when tumor tissue analysis is not available, while its performance is lacking in screening and low-grade disease.

The extent of genomic and transcriptional heterogeneity at the cellular level in BC remains mostly unknown. To determine the clinical value of the genomic and transcriptional landscape of bladder cancer in order to guide clinical management, it is necessary to conduct biomarker-driven clinical trials involving various stages of disease. In this regard, liquid biopsy will play a fundamental role in allowing a longitudinal design of the studies.

Future clinical trials or comparative studies between liquid biopsy and current diagnostic methods are necessary to improve the clinical application of this novel approach.

## Figures and Tables

**Figure 1 cancers-16-00803-f001:**
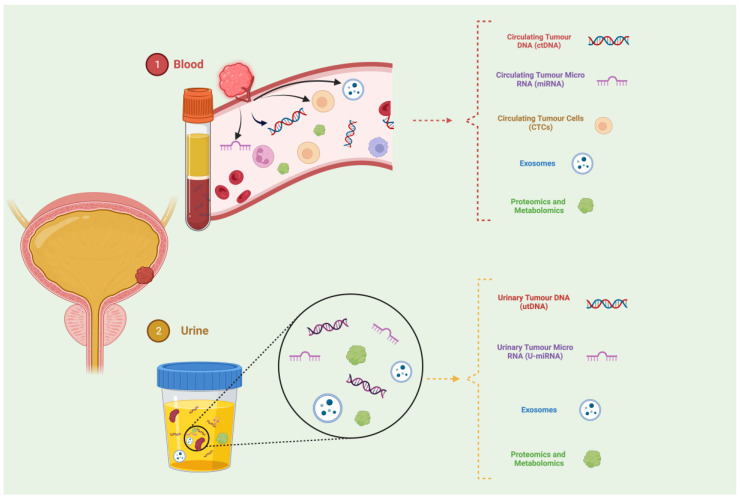
Overview of liquid samples and biomarkers in bladder cancer. Liquid biopsy is emerging as a promising method. It involves using peripheral blood or urine samples for the collection and examination of six tumor components: CTCs, ctDNA, ctRNA, exosomes, metabolomics, and proteomics. Peripheral blood and urine samples are then used to extract and examine tumor components.

**Figure 2 cancers-16-00803-f002:**
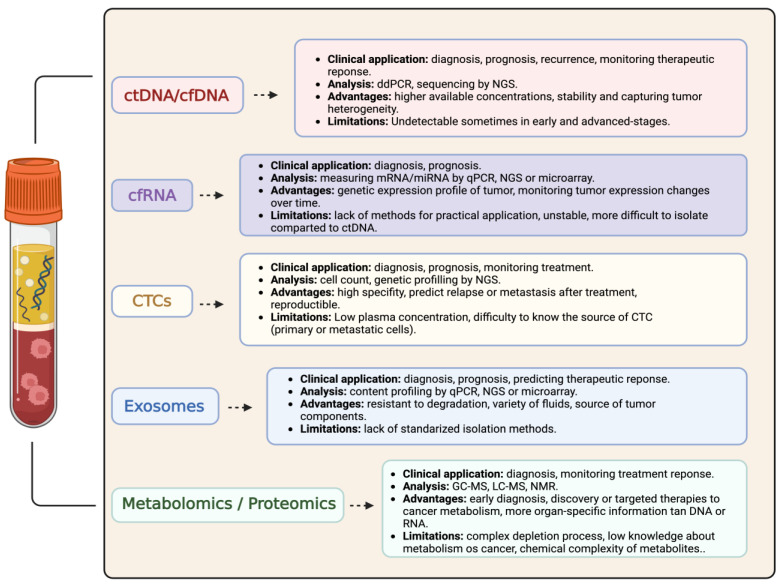
Major liquid biopsy methods in bladder cancer. Comparison of their clinical applications, laboratory analytical techniques, main advantages, and limitations.

**Table 1 cancers-16-00803-t001:** Summary of urinary biomarkers. Abbreviations—CEA: carcinoembryonic antigen; MAUB: mucin antigen of the urinary bladder; FDA: Food and Drug Administration; FISH: fluorescence in situ hybridization; CE: Conformitée Europeenne; NGS: next-generation sequencing; BS-Seq: bisulfite sequencing; PCR: polymerase chain reaction; SafeSeqS: Safe-Sequencing System; HTS: high-throughput sequencing; RT: real-time; MASO: multiplex allele-specific, oligonucleotide; qPCR: quantitative polymerase chain reaction; ELISA: enzyme-linked immunosorbent assay; POC: point of care.

Test	Variable	Biomarker	Assay	Clinical Application	Sensibility/Specificity	Reference
Urovysion	Chromosome 3-7-9-17	DNA/Sediment cells	FISH	Post BCG/early recurrence	69%/76%	[16,124,128]
Immunocyt	CEA, MAUB	Antigens and sulfated mucin glycoproteins (sediment cells)	Immunofluorescence cytology	LG-NMIBC diagnosis	77.5%/62.5%	[12,127]
BTA test	BTA	Protein	Immunochromatography + ELISA	Diagnosis and monitoring response	56%/85.7%	[114,115,116,117,118]
Uromark	Epigenetic alterations	Sediment cells/DNA	NGS + BS-Seq PCR	Predictive and monitoring treatment	95%/96%	[128,131,132]
NMP22 (Bladder Chek)	NMP22	Protein	ELISA + POC immunoassay	Early diagnosis and monitoring HG recurrence	59%/93%	[122,124]
Uromonitor	FGFR3, TERT, KRAS	DNA	PCR	Predictive (recurrence)	73.5%/93.2%	[13,128]
Uromutert	TERT	DNA	NGS PCR	Early diagnosis	87.1%/94.7%	[132,133,134]
Bladder Epicheck	DNA methylation	DNA	RT-PCR	Early diagnosis of HG-NMIBC	81%/83%	[135]
Uroseek	TERT, FGRF3, TP53, CDKN2A, ERB2, HRAS, PIK3CA, METH, BHL, MLL	DNA	SafeSeqS	Early diagnosis and monitoring response	95%/93%	[135,136,137]
Urodiag	FGFR3, HS3ST2, SEPT9, SLIT2	DNA	DNA methylation + MASO-PCR	Monitoring HG recurrence	95.5%/75.9%	[138]
CxBladder	CDK1, MDK, HOXA13, IGFBP5, CXCR2	mRNA	qPCR	Early diagnosis	82%/85%	[139,140]
Xpert BC	ABL1, UPK1B, CRH, ANXA10, IGF2	mRNA	RT-PCR	Exclude recurrence	76%/85%	[141]
ADXBLADDER	MCM	Protein	ELISA	Predictive (≳T1 disease)	73%/68.4%	[142]
CYFRA 21.1	Cytokeratin19	Protein	ELISA	Diagnosis	82%/80%	[143]
AssureMDx	FGFR3, TERT, HRAS, OTX1, ONECUT2, TWIST1	DNA	DNA methylation + PCR	Predictive (HG-NMIBC)	93%/86%	[144]
UBC Rapid	Cytokeratin 8 and 18	Protein	POC immunoassay	Predictive (Cis)	70.8%/61.4%	[145]
Oncuria	ANG, APOE,A1AT, CA9, IL8, MMP9, PAI1, SDC1, VEGF	Protein	Immunoassay	Diagnosis and follow-up	85%/81%	[146]

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
