# Peer review of "New Perspectives on the Role of Liquid Biopsy in Bladder Cancer: Applicability to Precision Medicine"

_cancers, 2024, doi:10.3390/cancers16040803_

Round 1
Reviewer 1 Report
Comments and Suggestions for Authors
The paper titled "New perspectives on the role of liquid biopsy in bladder cancer: applicability to precision medicine" is a comprehensive review focusing on the utilization of liquid biopsy in bladder cancer (BC) diagnostics and monitoring. The sections include an introduction, materials and methods, discussions on tumor heterogeneity in advanced BC, the role of liquid biopsy as a non-invasive test in BC, evaluation of commercial liquid biopsy kits, and a focus on AUTOTAXIN (ATX) with concluding remarks.
1. Innovative Techniques in Liquid Biopsy:
The review presents a comprehensive overview of liquid biopsy in bladder cancer, highlighting its role in understanding tumor heterogeneity and treatment resistance. It's commendable that the review discusses the diagnostic potential of circulating tumor cells (CTCs), circulating non-coding RNAs (ncRNAs), and cell-free DNA (cfDNA) in bladder cancer. This aligns with the current research trends and the importance of liquid biopsy in bladder cancer management.
2. Clinical Applications:
The review effectively highlights the clinical implications of liquid biopsy in bladder cancer. It emphasizes the need for non-invasive methodologies for early diagnosis and monitoring of bladder cancer, a key aspect of patient management.
Suggestions for Future Research:
1. Broader Literature Base:
While the current review provides valuable insights, future research could benefit from including a more extensive array of studies, especially those directly focusing on bladder cancer. This could provide a more comprehensive view of the field.
2. Technological Advancements:
Incorporate recent technological advancements in liquid biopsy for bladder cancer. For instance, a study by Kouba et al. (2020) discusses the current status and future directions of serum biomarkers in bladder cancer, emphasizing recent clinical studies and advances in methodology [Kouba et al., 2020, PMID: 31608720]. Including such studies could provide a more detailed perspective on technological progress.
Title and Abstract
- The title is well-crafted, clearly presenting the focus on liquid biopsy and its application in precision medicine for bladder cancer.
- The abstract succinctly summarizes the key aspects of the review, emphasizing the potential of liquid biopsy in the early diagnosis, detection of microscopic disease, and treatment response in BC.
Introduction
- Provides a solid background on bladder cancer, its prevalence, and current diagnostic methods.
- It would be beneficial to briefly discuss the limitations of current diagnostic methods to establish the need for alternative approaches like liquid biopsy more clearly.
Materials and Methods
- The methodological approach of conducting a narrative review based on literature search is well explained.
- Including selection criteria for the studies reviewed would enhance the credibility of the review process.
Tumor Heterogeneity in Advanced BC
- This section effectively addresses the challenges posed by tumor heterogeneity in BC.
- Incorporating recent studies that directly compare the outcomes of liquid biopsy with traditional methods in the context of tumor heterogeneity would add value.
Liquid Biopsy: A New Non-Invasive Monitoring Test in BC
- The section thoroughly explores the advancements and potential of liquid biopsy in BC.
- A discussion on the cost-effectiveness and accessibility of liquid biopsy compared to traditional methods would be a valuable addition.
Commercial Liquid Biopsy Kits for Bladder Cancer (BC) Screening and Follow-Up
- Provides a comprehensive overview of the available commercial kits, their functionalities, and their clinical applications.
- Comparing the performance and practicality of these kits in real-world clinical settings could provide a more applied perspective.
AUTOTAXIN (ATX): Future Perspectives of Liquid Biopsy in BC
- The focus on AUTOTAXIN is intriguing and presents a novel aspect of liquid biopsy research.
- It would be beneficial to discuss the current challenges in integrating ATX into clinical practice and potential strategies to overcome these.
Conclusions
- The conclusions aptly summarize the potential of liquid biopsy in revolutionizing BC diagnosis and treatment.
- Suggesting specific directions for future research, such as clinical trials or comparative studies with existing diagnostic methods, would be beneficial.
Overall, the paper is well-structured and informative, offering valuable insights into the emerging role of liquid biopsy in bladder cancer. The suggestions provided aim to further enhance its comprehensiveness and applicative value in the field of urology and oncology.
Comments on the Quality of English LanguageThe quality of English language in the reviewed document is generally good, with clear and coherent writing throughout. The text demonstrates a professional level of vocabulary and syntax appropriate for a scientific review article. There are a few instances where sentence structure could be slightly improved for better readability, but these do not detract significantly from the overall quality. The technical terms and concepts are well-explained, making the document accessible to readers familiar with the subject. The authors have done a commendable job in maintaining a consistent tone and style throughout the paper.
Reviewer 2 Report
Comments and Suggestions for Authors
The manuscript authored by Alberca-del Arco and colleagues provides a comprehensive and insightful review of recent developments and applications of liquid biopsy in the context of bladder cancer for precision medicine. The inclusion of an additional paragraph on autotaxin enriches the scope of the review. The authors adeptly present a table summarizing urinary biomarkers and tests, which serves as a valuable resource for readers seeking a quick reference for key information. The incorporation of two well-designed figures further strengthens the clarity and coherence of the text, effectively elucidating complex concepts and supporting the overall narrative.
The focus on liquid biopsy as a tool for precision medicine in bladder cancer aligns seamlessly with the objectives of the special issue.
I recommend the acceptance of this well-structured manuscript for publication in the special issue of Cancers.
Reviewer 3 Report
Comments and Suggestions for Authors
This is comprehensive study dedicated to very important field in cancer biology. It is very critical to organize data about current and developing biomarkers, especially related to liquid biopsy. Even more important is relation to bladder cancer. This field is less developed than other oncology diseases. Review is generally well organized, adequately written and references are correctly presented.
However, I would like to add some notes that to my mind may improve the manuscript.
1. 4.1.2 line 294 "we describe" sounds that authors cite their own paper.
2. In this chapter (RNA) authors dedicated a lot of space to several type of RNA but not even mentioned circular RNA that is definitely becoming very important biomarker (including BC).
3. Some phrases are confusing -line 364 Why "however"?
4. Line 373 This paragraph is not very clear or necessary.
5. Line 386 "Their evaluation enables us to develop...' this citation of not author's paper.
6. Please rephrase line 406-407
7. text 410-425 too long
8. Line 474 fix word "profiling"
9. the last paragraph (line 488-495) will better fit in the next chapter
10. I would make this chapter (5.1) shorter.
11. line 606 to my mind "diluted" is not good word here
12. line 635 "however" not in the place here
13. ATX story is very intriguing but not directly related to the current review.
Reviewer 4 Report
Comments and Suggestions for Authors
The manuscript focuses on the promising role of liquid biopsy derived analytes in clinical management of bladder cancer patients represents a technically correct and timely relevant manuscript available for the publication on this journal after minor suggestions
- In the text, please, could the authors underline the clinical issues met in the stratification of bladder cancer patients?
- In the text, please, could the authors group the well explained technical approaches in accordance withe their clinical application? (diagnosis, prognosis, prediction)
- In the table 1, please, could the authors review column 4 (assay) reporting assay type and in a separate column proper name of adopted assay?
- Please, could the authors report gene in italics in accordance with standardized nomenclature system?
- Please, could the authors review native english style and form
Comments on the Quality of English LanguagePlease, could the authors review native english style and form
